**PLOS** NEGLECTED TROPICAL DISEASES

# Challenges Experienced and Observed during the Implementation of Leprosy Strategies, Sidama Region, Southern Ethiopia: An inductive thematic analysis of qualitative study among health professionals who working with leprosy programs

**Kebede Tefera Betru**[1]*, **Thuledi Makua**[2]

**1** Hawassa University, College of Medicine and Health Sciences, School of Public Health, Hawassa, Ethiopia, **2** University of South Africa, College of Human Sciences, Pretoria, South Africa

\* ktefera2015@gmail.com

**Data Availability Statement:** All relevant data are in the manuscript.

## Abstract

### Background

Prompt diagnosis and treatment of leprosy are crucial for preventing the disease's spread as well as for avoiding negative medical and social effects and reducing the disease's burden. The likelihood of nerve damage and subsequent disability rises as the length of the diagnostic delay. We aimed to explore the challenges of health professionals faced regarding their involvement in early leprosy case detection strategies.

### Methods

The study employed a qualitative, descriptive and phenomenological explorative research design to answer the research questions. By the use of non-probability purposive sampling, research participants were identified. During the study, in-depth interviews were conducted to gather information regarding the experiences of health workers (medical doctors, public health officers, clinical nurses, health centre heads and regional and Woreda district health office technical and programme experts) and health extension workers. To analyse the qualitative data, inductive thematic analysis techniques were used. For analysis, open code software version 4.0 was used. The data transcription, coding, display, reduction (theme) and interpretation of the discovered results were the processes undertaken for the analysis.

### Result

The findings of the study revealed that leprosy prevention and control programmes are still problematic. Themes that emerged from the data gleaned from the health workers included: lack of the existence of practice-oriented training, Integration of TB and leprosy training, lack of focus or other competing health priorities, Inadequate supportive supervision of health

**Funding:** This work was supported by Hawassa University to KT. The funder had no role in study design, data collection and analysis, decision to publish, or preparation of the manuscript.

**Competing interests:** The authors have declared that no competing interests exist.

facilities, Multiple tasks for health workers, poor coordination and communications, lack of motivation in health workers, disruption in treatment, and Importance of training related to leprosy.

## Conclusion

Strengthening comprehensive leprosy training for health workers, carrying out efficient and thorough contact tracing, enhancing monitoring, supervision, assessment and surveillance, boosting managerial skills, lobbying political commitment, and motivating healthcare workers may help in early detection of leprosy cases strategies.

## Author summary

Leprosy is a chronic, infectious and contagious disease caused by the M. leprae bacterium. Skin lesions, loss of sensation, nerve damage, muscle weakness, and deformities are common characteristics of leprosy. Prompt diagnosis and treatment of leprosy are crucial for preventing the disease's spread as well as for avoiding negative medical and social effects and reducing the disease's burden. However the likelihood of nerve damage and subsequent disability rises as the length of the diagnostic delay increases. In this study, we identified that health professionals face many challenges to implement early detection of leprosy cases strategies like lack of the existence of practice-oriented training, Integration of TB and leprosy training, Inadequate supportive supervision of health facilities, Multiple tasks for health workers, Poor coordination and communications, lack of motivation in health workers, Disruption in treatment, and Importance of training related to leprosy outlined. The effective implementation of these strategies will be required for a sustainable reduction and achieve zero autochthonous cases and comply with the WHO's Leprosy Strategy by 2035.

## Background

Leprosy, also known as Hansen's disease, is one of the oldest known human diseases, dating back to ancient civilizations. It primarily affects the nerves, skin, and mucous membranes, leading to disfiguring skin sores, loss of sensation, and nerve damage. It has affected millions of people throughout history and has posed significant socio-cultural challenges due to its disfiguring symptoms and the associated stigma. The causative agent of leprosy, Mycobacterium leprae, was first identified by Gerhard Armauer Hansen in 1873 using microscope observations of bacteria in skin biopsies of leprosy patients [1, 2]. It primarily spreads by droplets that are emitted from the nose and mouth during breathing; its transmission is at a high level with patients who are infectious yet untreated. In contrast to other illnesses with a high mortality rate, leprosy is a chronic illness that slowly worsens over time and sometimes leaves behind residual disability. It is estimated that 3–4 million people are living with visible impairments or deformities due to leprosy [3]. Skin lesions, loss of sensation, nerve damage, muscle weakness, and deformities are common characteristics of leprosy. Since then, various diagnostic methods have been developed to identify leprosy, including slit-skin smears, histopathology, and molecular techniques such as polymerase chain reaction (PCR). These methods have enhanced our

understanding of leprosy and improved early diagnosis and treatment outcomes, contributing to the control of the disease worldwide [1, 2].

The diagnosis of leprosy in current practice is based on the presence of at least one of the three cardinal signs: (i) definite loss of sensation in a pale (hypopigmented) or reddish skin patch; (ii) thickened or enlarged peripheral nerve with loss of sensation and/or weakness of the muscles supplied by that nerve; or (iii) presence of acid-fast bacilli in a slit-skin smear. Slit-skin smears are positive only in MB leprosy (i.e. any positive slit skin smear is classified as MB irrespective of the number of patches and/or nerve involvement) [4].

According to the WHO classification, leprosy patients are categorized depending on the number of leprosy skin lesions and nerve involvement. The classification also helps on choosing the treatment regimen and predicts the future risk of complication: Paucibacillary (PB) Leprosy (a) One to five leprosy skin lesions, (b) Only one nerve trunk enlarged. Multibacillary (MB) Leprosy (a) Six or more skin lesions, (b) Less than six skinlesions, which have a positive slit skin, smear result or if there is involvement (enlargement) of more than one nerve. Additionally, leprosy patients that are difficult to categorize should be considered multi-bacillary cases and handled as such. Pure neural leprosy patients should also be categorized and managed as MB cases [5]

Leprosy was vertically treated in Ethiopia's healthcare system until 2001 at hospitals with a focus on the disease. By the end of 2001, the leprosy control program had been effectively incorporated into the provision of general healthcare services, ensuring that patients receive an early diagnosis and successfully complete the multidrug course (MDT) with no concomitant disability [6].

Due to the chronic impairment and social stigma it causes, leprosy has been a concern for public health for many centuries. Early detection and prompt treatment with multidrug therapy (MDT), the basic tenets of leprosy control, led to a significant reduction in the burden of leprosy by the early 2000s. During 2021, 140 594 new cases were reported globally, for a case detection rate of 17.83 per million population. The rate of detection of new cases increased by 10.2% as compared with 2020 (128 405)New cases in children indicate recent transmission, and 9052 new child cases were reported globally, with a corresponding rate of 4.5 per million child population. An increase in the number of child cases (4.7%) was observed in 2021 when compared with 2020 (8642) [7]. The focus on delivering diagnostic and treatment services that are fairly dispersed, reasonably priced, easily available, acceptable, and of a high standard will continue to be a key component of leprosy control. However, the ability of health professionals to identify the early signs and symptoms of leprosy might pose a barrier to the diagnostic and treatment services [8–10].

The lack of highly specialized expertise among general health staff working in district and zonal hospitals is likely caused by the fact that the majority of these staff members were primarily focused on treating other illnesses and were therefore unmotivated to become knowledgeable about leprosy. Given that the majority of leprosy cases are seen and treated in health centers, the relatively high number of general health professionals with a positive attitude among them offers a solid potential to improve through a focus on in-service training [11].

According to the survey, most healthcare professionals were inexperienced in identifying leprosy's early symptoms, reaction, and treatment. The vast majority of them did not know how to conduct the diagnostic voluntary muscle tests or sensory tests for leprosy.. Even though the prevalence of leprosy has significantly decreased as a result of the deployment of MDT, there are still ongoing occurrences of new cases being discovered and transmission is occurring in select highland areas of Ethiopia. One of the most significant markers of late detection in healthcare facilities, according to 2019, reported a total of 3426 leprosy cases. Among the total registered cases 2957 (86.3%) were new and 368 (10.7%) were registered as relapse [12].

The majority of the time, when cases of mb exceed 50% of new cases, they are frequently linked to a delayed diagnosis. [13]. Grade II disability which means visible deformity or impairment, of hands and feet, including claw hand, drop foot, and visible damage to hands or feet and eyes which means significantly restricts normal work, activities, or social life [14].. Prompt diagnosis and treatment of leprosy are crucial for preventing the disease's spread as well as for avoiding negative medical and social effects and reducing the disease's burden. The likelihood of nerve damage and subsequent disability rises as the length of the diagnostic delay; for this reason, G2D is widely used as a proxy indication for the diagnostic delay [15].

Traditional or religious beliefs about leprosy include that it is a curse from God or an ancestor, retribution from sins, or the result of witchcraft. There are several myths and misconceptions surrounding leprosy in Ethiopian culture. One prominent misconception is that leprosy can be transmitted by touch or casual contact, leading to the fear of infection among community members. This misconception has further contributed to the stigmatization and discrimination faced by individuals affected by leprosy. The stigma associated with leprosy has been sustained and increased by disability, disfigurement, and the disease's stigma, which causes those who are afflicted to withdraw from society, hide their status, and delay receiving treatment [16, 17].

One of the major contributing factors towards the late diagnosis of leprosy is communities' lack of knowledge regarding leprosy ultimately leading to increased likelihood of physical disability [18, 19]. The low prevalence of leprosy is associated with a fear of the loss of leprosy-specific skills within the healthcare services that could result in considerable delay in the diagnosis and treatment of the disease [20].

Early diagnosis of leprosy poses several challenges, partly due to the complex nature of the disease and the lack of specific diagnostic tools. However, advances in technology have led to the development of more sophisticated diagnostic tests that aid in the early detection of leprosy. Here are some of the challenges and the corresponding references regarding sophisticated diagnostic tests: Leprosy diagnosis depends on clinical symptoms and microscopic examination of skin smears. However, these methods lack specificity and sensitivity, especially in early cases [21]. The clinical symptoms of leprosy may take months or even years to manifest after infection, contributing to delayed diagnosis and treatment initiation [22].

Subclinical cases are individuals who are infected with M. leprae but do not exhibit any visible signs or symptoms of the disease, making early detection problematic. The availability and accessibility of sophisticated diagnostic tests, such as PCR or molecular tests, may be limited in resource-limited settings, hampering early diagnosis efforts [23]. So, a multidisciplinary team approach is crucial for early diagnosis in leprosy due to the challenges posed by sophisticated diagnostic tests. The expertise of clinicians, pathologists, microbiologists, rehabilitation specialists, social workers, counselors, and health educators ensures a comprehensive assessment, accurate diagnosis, and timely intervention. This holistic approach not only aids in early detection but also addresses the psychosocial and rehabilitative aspects of leprosy, ultimately improving patient outcomes.

A successful leprosy care and control programme within the general healthcare services at the PHC level is highly dependent upon the HWs having adequate knowledge of, and practical training on, leprosy [24]. In this light, we explored the challenge of the early diagnosis of leprosy cases by in-depth interviewing health professionals in endemic part of the country.

## Methods

### Ethics statement

Before conducting the interview, ethical approval was obtained from University of South Africa and Hawassa University Ethical Review Committee (Reference number: HSHDC/911/

2019, IRB/275/12) and permission obtained from the District Health Office. The purpose of the study was clearly described for each study participant. Verbal consent was taken from each participant. Because verbal consent offers flexibility, allowing for adaptations or clarifications in the consent process as required Confidentiality of study participants was kept using code instead of identifying them with their name. All the information was coded for anonymity and only the investigators have access to the data. The participants were involved voluntarily and they were informed of their rights to participate or withdraw from the study at any time.

## Type of study and site of research

A descriptive qualitative design with a phenomenological approach used in this study. The phenomenological approach in health professional study refers to a methodology that focuses on the individual's lived experiences, perceptions, and interpretation of a particular phenomenon related to healthcare. This approach aims to understand and describe the subjective experience of individuals, rather than analysing their behaviour or objective data.. The study was carried out in the Bursa District, Sidama Region, South Ethiopia. The distance between the district and Addis Abeba, the capital, is 214 kilometres. Bursa is bordered on the south by Hula, on the west by Aleta Wendo, on the northwest by Wensho, on the northeast by Arbegona, and on the southeast by Bona Zuria Based on the 2007 Census conducted by the CSA, this district has a total population of 103,631, of whom 51,731 are men and 51,900 women; 2,304 or 2.22% of its population are urban dwellers. The majority of the inhabitants were Protestants, with 88.63% of the population reporting that belief, 6.25% observed traditional religions, 2.18% were Catholic, and 1.77% were Muslim. [25]. The language that is most commonly used is Sidamigna. In 2020, there are 22 public health facilities (3 health centres and 19 health posts) in the Bursa Woreda. According to the report of Woreda health bureau statics the major health problems in the study area were communicable diseases including Leprosy. The researcher carried out the study on all health facilities providing leprosy diagnosis and treatment.

The study period was from March 12 to April 9, 2019,

## Study participants and sampling techniques

In this study the population consisted of health care workers(health centre Head chief executive officers (CEO), district health office TB/ leprosy experts, regional health bureau level TB/ Leprosy experts, health centre OPD health workers, health centre TB/leprosy focal and community health extension workers(HEWs) providing leprosy prevention and control program in the selected area. The researcher obtained staff lists at the study region health bureau and purposively selected who were likely to be sufficiently knowledgeable and who had had experience in the management of leprosy cases and programmes by virtue of being able to meet the selection and inclusion criteria set by the researchers were the study subjects. The purposive sampling was used to select participants who could provide information-rich data and study area for the study depending on their goals [26]. The researchers selected a total of 23 participants (18 health workers and 5 health extension workers providing leprosy prevention and control program information as participants.

The participants were involved voluntarily and were given information about their rights to join or discontinue their involvement in the study whenever they desired In qualitative research, the sample size is typically determined by the information required. The number of study participants is known as the sample size and, according to Creswell and Poth and Polit and Beck ([27, 28]) the size of the study participants is determined by the information needs of the research and guided by the principles of data saturation.

The point of saturation in qualitative studies refers to the moment when the sample size is sufficient to capture all relevant themes, insights, and information related to the research topic. It signifies that no additional data is needed as the collected information consistently fails to uncover new perspectives or contribute to enhancing the understanding of the research topic. Essentially, saturation indicates that a comprehensive comprehension of the subject matter has been attained through the available data [29].

The researcher stops sampling when no new information is revealed. Participants were invited for in-depth interviews at their convenient time and place.

## Method of data collection and analysis

**Data collection.** Individual, face-to-face (in-depth), unstructured interviews were used to gather the data. Unstructured interviews frequently begin with an open-ended question about the topic of the study, with the participant's comments influencing the questions that follow [30]. The level of in-depth investigation was raised through probing and follow-up questions. The interview continued until information saturation is reached. The in-depth interviews were taped and note taken by the interviewers. Participants were interviewed at their respective work areas within one week. To protect the participant's privacy and confidentiality, the interviews were held in a private space. The response of each participant was audio-recorded and notes were taken at each interview. The interviews were ranged from 45-60minutes. Before conducting the main study, a pilot study was carried out in the corresponding health center office, health post, medical physician, health officer, nurses, and health extension workers (HEW) were selected. The interviewers were chosen among health professionals who possessed extensive knowledge and experience in neglected tropical diseases. Moreover, they exhibited proficiency in interview techniques and possessed extensive knowledge of the local environment, supplemented by comprehensive training. The interviews were transcribed verbatim after being tape recorded with the interviewees' consent. The audiotapes allowed the researchers to frequently review the data for confirmation when reviewing later.

## Measures for ensuring trustworthiness

**Credibility.** The basis of any claim to judging the trustworthiness of qualitative research criteria. To increase the credibility of the qualitative study, the researcher applied different sources to conclude what constitutes the truth. Therefore, the researcher repeatedly interviewed participants until new data were not forthcoming.

**Dependability.** It demonstrates the reliability of information throughout time and environments. To enhance the dependability of this research, the researcher undertook an inquiry audit. The collected data were scrutinised against the available support data during this procedure. The research inquiry and the supporting documents were made available to an external auditor (supervisor) who carried out the audit.

**Conformability.** The capacity of data to be accurate, relevant or meaningfully consistent. By allowing other researchers to see the audit trail before the trial began; the researcher was able to increase conformability. Before starting the trial, the researcher was required to make any necessary corrections that were discovered by the other researchers.

**Transferability.** The capability to apply the findings of the study to other people or circumstances is what transferability. To improves transferability; the researcher used thick description, which was rich and thorough in the explanations and discussion of the concepts under investigation. Through this, other researchers can relate to or use the results of this study in other similar research settings or contexts.

**Data analysis.** Inductive thematic analysis was employed to analyse data. The inductive approach is particularly useful in exploratory research, where the goal is to explore and understand a phenomenon or topic in-depth when there is limited existing knowledge. Data collection and data analysis were performed simultaneously and before conducting the next data collection importantly emerged ideas were identified by listening audio-recorded material and reading field notes. Also, data saturation was ensured through this process. The data saturation of this study was identified through ongoing data comparison. Verbatim transcription was done by listening audio-recorded material and translated from Amharic to English in support of interviews and researchers, and also checked for completeness and consistency. Reading and re-reading of the data were carried out to become familiarize with data. Then, the principal investigator and the research assistant coded the transcription line by line on OpenCode 4.0 Umeå software (ICT Services and System Development and Division of Epidemiology and Global Health, Umea University, Sweden, 2015)starting from the richest data. Inter-coder consistency was checked, the codebook manual was developed, and then, the principal investigator coded the whole data again and again by refining the codebook. Throughout the coding system, code consistency was checked throughout the coding process by reading and re-reading and re-coding. Then, codes were clustered into categories and themes were developed by connecting related categories. Finally, the report was organized based on major themes, categories, and quotations taken from participants.

## Results

### Socio-demographic characteristics of study participants

All relevant levels of the health system—including the community level, the facility level, the district health office, the regional health bureau and others—provided participants for this study. Everybody was qualified, barring the community-level health professional and all had degrees or higher. Most of the participants had worked on initiatives connected to leprosy for more than five years. The majority of participants at the time of the interview either oversaw a leprosy-related office or directed initiatives to combat leprosy and tuberculosis. Table 1 demographic characteristics and occupational information of the participants in the study.

Twelve key informant health workers and health extension workers interviews with in the study villages were conducted. The in-depth and key-informant interview findings were combined and organized based on the following major themes.

**Table 1. Demographic characteristics and occupational information of participants in the study.**

| Participant code | Age | Sex | Professional sub-theme | Position | Length of experience in years in TB/Leprosy |
|---|---|---|---|---|---|
| P1 | 38 | M | Master's in Public Health | South Region Global Fund TB-Leprosy Officer | 12 |
| P2 | 46 | M | Master's in Public Health | Sidama Region Health Bureau TB/Leprosy Dept. Head | Over 15 |
| P3 | 43 | F | Masters in Epidemiology | South Region Health Bureau TB/Leprosy Officer | 9 |
| P4 | 50 | F | Master's in Public Health | South Region Health Bureau TB/Leprosy Dept. Head | 14 |
| P5 | 29 | M | Health Officer | Bursa District Health Office CDC Dept. Head | 8 |
| P6 | 31 | M | Health Officer | Bursa District Health Office CDC officer | 7 |
| P7 | 37 | M | Health Officer | Bursa Health Centre Head | 6 |
| P8 | 40 | M | Health Officer | Bureau TB/Leprosy Dept. Head | 8 |
| P9 | 33 | F | Health Extension Worker | Health Post Community Health worker | 6 |
| P10 | 37 | F | Health Extension Worker | Health Post Community Health worker | 7 |
| P11 | 27 | F | Medical Doctor | Bursa Health Centre OPD | 3 |
| P12 | 39 | M | Health Officer | Bursa Health Centre OPD | 5 |

## Challenges experienced and observed during the implementation of leprosy strategies

The conversations with the participants made clear that numerous obstacles limit the implementation of leprosy solutions and put the health and safety of people affected by leprosy at risk. Participants believed they were falling short of their objectives in terms of early leprosy case detection and providing proper medical care due to these difficulties. Key participants explained several challenges and provided suggestions to all levels of health administration presented in the following themes.

### Lack of the existence of practice-oriented training

All participants believed that theoretical knowledge was good but needed to be strengthened by practical experiential learning. This exposition relates to the integration and implementation of theory and practice for deep engagement in the learning process.

during *training, we were never allocated in the leprosy centre to be able to practice the implementation of assessment of leprosy cases and managements. So . . . the training programme has a practical gap which health professional needed to fill.* [P12]

### Integration of TB and leprosy training

Participants agreed that integrating TB and leprosy training is not advised due to a lack of fair time allocation, the trainer's lack of experience with interdisciplinary planning and implicitly limited familiarity with the specific practices, the absence of some good practice examples and the scarcity of efficient models from the current practice of leprosy management.

*Leprosy-based training is part of comprehensive tuberculosis training. Leprosy-related training is provided at the final day. The trainees do not acquire the required knowledge though it requires a practical training. First, they should learn about the lesion identification using real cases or videos. Physical diagnosis needs a one-day practical training. For instance, they should learn about assessment of nerve enlargement sample taking from a lesion. Therefore, the training that is integrated with tuberculosis training is ineffective . . . Health providers should take the training especially; it should be provided for those involved in the diagnosis of patients . . . Likewise, pre-service training should address leprosy. All health professionals should learn about leprosy client. Therefore, training should be provided for every health professional.* [P1]

### Lack of focus or other competing health priorities

A key participant reported that lack of focus or other competing health priorities is one of the key challenges in controlling leprosy.

*It is all related to focus given. No attention is given to leprosy at all quarters of government. For example, are we seeing awareness creation of leprosy TV or radio? No, but it has spread in the community. Since TB REACH project phase out, it has been dropped out. There has to be a responsible and supporting body; there is urgent need to find supporting body like NGOs as before. If things go this way it will be worse than COVID-19.* [P6]

Another participant shared the above idea and said:

*There are certain diseases that cause fear. For instance, when measles occur, posters are posted everywhere. On the other hand, low attention is given for leprosy. As for leprosy, we are not working at the kebele [community] and health facility level. People come for care by themselves. Health professionals should create awareness on the severity of the disease. I have seen a case with a family history of leprosy. Therefore, the community should know that it might be*

*transmitted by contact with the case. The health professionals should also know this. Therefore, health professionals who are treating skin infection case should always suspect leprosy.* [*P8*]

Another participant held this belief:

*I think leprosy control programme is less important than any other programme that is why a reduced amount of attention given for leprosy control programmes.* [*P5*]

## Inadequate supportive supervision of health facilities

All participants believed that the regional/zonal/woreda health bureau conducted supportive supervision at health facilities (hospitals, health centres and rural health posts) to enhance the health practitioners' ability to provide healthcare services. However, for a variety of reasons, this supporting supervision was not conducted regularly. Regional/zonal and woreda health bureaus did not monitor regular basic supervision at different levels of the health sector to strengthen the identified gaps. A Bursa head said:

*As a general rule, we must visit the health post once a week. However, because of various tasks, a lack of personnel and supplies (a motorcycle), we travel on an irregular basis and are unable to perform the policy's recommended level of monitoring.* [P8]

The head of the health centre shared his experience in supervision and monitoring.

*We know to perform the supervision and monitoring. But there is no continuous supervision and monitoring as well as no specific budget allocation.* [P7]

Some participants held the opinion that the evaluation procedure is poor because TB/leprosy units have not received adequate input.

*. . . In order for the TB/leprosy units in hospitals, health centres and health posts to understand the status of leprosy incidence in the community and avoid not taking it seriously, they need to get periodic feedback.* [P11]

## Multiple tasks for health workers

Several tasks for health workers may cause a decreased emphasis on the services of leprosy such as prevention of disability and transmission. A regional leprosy focal person explained that facilities and community-level detection of leprosy cases are doing less.

*The TB/leprosy focal health personnel, in addition to their main tasks, they have other different health facility assignments, responsibilities so these an excessive work burden may lead to less attention for the services.* [P2]

A participant emphasised the workloads of HEWs rather than their duty commitment as reasons for poor performance.

*. . . Most the time health extension workers busy of other activities in the community such as tax collection and so on . . . to implement 16 health packages through the community level with two health extension workers, it is too challenging.* [P3]

Another participant explained that poor integration among sectors was a challenge.

*. . . Bureaucracy and weakness in inter-section coordination causes an excessive work burden. These may lead to less attention for the services of leprosy control programme. Generally, these were made unsuccessful in case management and cases finding.* [P4]

The detrimental effect of limited resources, specifically human and material resources, on the execution of services was cited as another challenge.

*. . . there is shortage of staff against many important duties at health facilities, these may lead to less attention at essential areas of the strategies . . . these may follow low performance.* [P12]

### Poor coordination and communications

According to one member, coordination and communication within healthcare facilities are poor.

*TB/leprosy coordinating officer has a weak communication with health extension workers those works in community health. They can't give timely feedback.* [P2]

*With respect to proper intra-section collaboration and inter-section coordinating engagement, we don't have a strong working relationship among us.* [P11]

**4.3.1.4.7 Lack of motivation in health workers.** Participants agree that appropriate motivation in the health sector can enhance the performance of the health system; however, appropriate motivation was not consistently practised for a variety of reasons. Employees who lack the proper motivation may be unable to cooperate appropriately and effectively. Several explanations have been offered in this regard.

*Mostly focal personnel are discouraged from identifying patients by the prolonged treatment follow-up period, contact tracing without allowances and potential for contracting disease.* [P8]

### Disruption in treatment

A facility head mentioned that one of the challenges they face relates to clients on treatment who stop taking the drugs when they feel healthy. In some instances, an interruption in the medication supply chain leads to missed doses. The interviewee also expressed concern that clients' support needs go beyond medication; they need mental health and financial support.

*Many come at stage one or two, but the main problem is that some patients miss the follow-up when they feel healthy and again they come back when they feel ill . . . The other though we face medication shortage rarely; it could lead patients to miss dose they supposed to take . . .Those clients who missed doses return when their condition worsens. Some patients are geriatrics and have mental problems. I know a female and male old patient with this problem. Those patients try to find a support using various means rather than focusing on the treatment of the disease. Some may seek for money support.* [P7]

Distance and transportation costs to leprosy treatment facilities represented another major challenge as a medical doctor explained:

*Most of the time leprosy-affected people to be economically dependent or weak is that the stigma attached to the disease, due to this reason delayed of early treatments, interruption of monthly refill drugs and follow-up or appointment are affected most the time.* [P11]

### Importance of training related to leprosy outlined

Almost all participants' findings under suggestions for improving the health workers' performance in respect of knowledge and skills for leprosy prevention control and management indicated that they should be receiving regular and updated training.

*. . . If once health professional have inadequate knowledge and skills for leprosy prevention, control and management, this may lead to missed early case detection of new leprosy cases and appropriate dealing with leprosy so . . . strongly recommend that all health professionals should receive training with the trainers have relevant working experience in the field within the following areas clinical features and diagnosis of leprosy, grading of disability, classification and treatment, algorithms for managing persons with leprosy in a clinical setting, screening of contacts (household/close contacts, community contacts), self-care with practical attachments of clinical area . . .* [P7]

*. . . Regular refresher training for healthcare professionals is necessary to keep their knowledge and abilities current and to ensure that patients receive high-quality care and the community . . . especially focuses on pre-treatment counselling, chemoprophylaxis, extent of contact tracing,*

*disability prevention and medical rehabilitation, leprosy reaction and management, management of MDT side effects, treatment in special conditions, referral of leprosy patients for special care, counselling of the index patients, counselling of contacts, complications of leprosy and their management, supporting with audio-visual aids.* [P11]

One of the key factors in the early detection and prevention of leprosy, especially in the areas of mode of transmission, sign and symptom, prevention control and management, stigma and treatment supporter, is increasing community-based leprosy prevention and care awareness through meetings, community mobilisation activities and trainings. [P2]

## Discussion

This is the first qualitative study amongst health care providers related to leprosy cases in Sidama region, south Ethiopia. While leprosy health care workers agreed that level of stigma was decreasing, both groups believed that it was still possible and was associated with visible physical deformities and societal stereotypes and affected many aspects of leprosy affected persons' lives. This study explored a variety of challenges that health professional experienced and observed during the implementation of leprosy strategies.

In this study, lack of the existence of practice-oriented training was identified as the main challenge for early detection of leprosy cases. This finding is consistent with other prior studies conducted in Ethiopia, in which almost 86% of the health workers had low-level knowledge were identified, which could be attributed to little attention given during formal training, lack of practice after training and low number of leprosy cases for practice per site [11]. The finding of this study is also comparable with a Malaysian study, HCW that reported seeing fewer cases at work were also found to have less knowledge about diagnosis and management of leprosy which reduced their ability to detect cases. The apparent lack of exposure to leprosy therefore may be a result of patients presenting with leprosy signs being missed and/or misdiagnosed by HCW than a true absence of cases in those areas. That would be a challenge in the fight against leprosy as undiagnosed cases might increase chances of disease transmission in their communities [31]. Healthcare professionals who have had leprosy training and have access to reference materials on the disease demonstrated greater knowledge and skill [32].

In this study, Integration of TB and leprosy training affects the appropriate utilization of early detection of leprosy cases. The absence of some good practice examples and the lack of efficient models from the current practice of leprosy management were all areas where the participants agreed that the integration of TB and leprosy training is not advised. Other points of the agreement included the trainers' lack of experience with interdisciplinary planning and, implicitly, relatively superficial knowledge of the specific practice [33]. Meanwhile the ancestors of Mycobacterium leprae and Mycobacterium tuberculosis diverged from a common ancestor millions of years ago, leading to distinct evolutionary paths and differences in the diseases they cause and the body systems they affect. Leprosy primarily affects the skin, peripheral nerves, and mucous membranes. It presents in different forms ranging from tuberculoid (localized, milder) to lepromatous (widespread, severe). Tuberculosis, on the other hand, primarily affects the lungs but can also spread to multiple body systems [34, 35].

In this study lack of focus or other competing health priorities were identified as the main challenge for early detection of leprosy cases. This finding is consistent with study conducted in Ethiopia majority of health professionals focusing on treating other illnesses and being uninspired to learn more about leprosy, the general health staffs at district and Zonal hospitals lack high levels of experience [11]. Similarly other study showed that the influence of the implementation degree of the effects observed, by component, was convergent and divergent between some indicators, especially among the unsatisfactory: low proportion of contacts

examined, treatment dropout, limited standardization of patient care flow, and inadequate resolution of issues with management support [36].

This study also revealed that inadequate supportive supervision of health facilities was one of the challenges of affecting the program. Similar findings were found in another study, which found that the influence of the implementation degree of the effects observed, by component, was convergent and divergent between some indicators, particularly among the unsatisfactory: low proportion of contacts examined, treatment dropout, limited standardisation of patient care flow and insufficient resolution of issues with management support [36].

The assignment of various jobs at healthcare institutions as a result of a lack of healthcare staff may result in patients receiving less attention due to overload. Due to a shortage of health workers, moving from one job to another negatively impacts the previous position held in that facility [37].

This study also revealed that Poor coordination and communications were another challenge of implementation of leprosy prevention strategies. Low attention to leprosy control initiatives is a concern in the fight against leprosy. This finding is consistent with study conducted in Ethiopia; high endemic states usually have subpar administration and a dearth of educated people [36]. Similar findings from another study show that some indicators, especially those that were unsatisfactory—a low proportion of contacts examined, treatment dropout, a limited standardisation of patient care flow and insufficient resolution of management support issues—were influenced by the degree of implementation of the effects observed by component [36].

The study explored that multiple tasks for health workers were challenging to carrying out the leprosy prevention and control program. Similar findings from another study show that the assignment of various jobs at healthcare institutions as a result of a lack of healthcare staff may result in patients receiving less attention due to overload. Due to a shortage of health workers, moving from one job to another negatively impacts the previous position held in that facility [37].

This study identified that lack of motivation among health workers and health extension workers employee disengagement and poor service quality can be caused by low morale in the workplace. Performance in work is greatly influenced by motivation. Intrinsic and extrinsic variables both inspire healthcare personnel. The effective work performance of health personnel depends on their understanding of important financial and nonfinancial motivational elements. The level of motivation in an employee affects how well the organization performs; as a result, a motivated employee is more likely to be physically healthy, mentally stable, and socially aware, which will enable them to do their jobs as required. Resources are channelled to achieve the most output from employees because managers and administrators understand what motivates people to give effective work performances [38, 39].

This study demonstrates that inadequate coordination across numerous healthcare systems, which results in barriers to care, subpar technical quality, interruptions in service, and inefficient resource usage, is one of the major challenges to achieving effective management of leprosy prevention.

Many healthcare systems throughout the world believe that one of the biggest barriers to achieving effective healthcare is a lack of coordination among healthcare levels, which causes problems with access to care, low technical quality, discontinuity of care, and inefficient resource usage [40, 41].

In this study, the importance of training related to leprosy was identified. Similarly Oman MoH regular training sessions are held for staff members at primary healthcare (PHC) and dermatology clinic levels to ensure that they have a uniform understanding of how to suspect and diagnose leprosy cases as well as to remind them of the necessity of promptly reporting

leprosy cases [42]. This finding is consistent with study conducted in Ethiopia, Leprosy health education and ongoing training should be focused at the pre-service and in-service levels to support the development of health workers' skills, knowledge and attitudes [11].

The study explore the disruption in treatment was one of the challenges that hindering the program of leprosy control. This finding is consistent with study conducted in Brazil. Poor adherence has detrimental implications such as incomplete healing, ongoing infection sources, the spread of new susceptible and antibiotic-resistant strains and the possibility of developing disability or deformity [43]. A lower defaulter rate would be made possible by health workers regularly following up and educating patients and their caregivers [44, 45].

The key strategies that recommended from the analysis of the qualitative data are as follows: The strategies include implementing and strengthening effective community awareness-building efforts, maintaining expertise in leprosy, strengthening comprehensive leprosy training for health workers, carrying out efficient and thorough contact tracing, enhancing monitoring, supervision, assessment and surveillance, boosting managerial skills, lobbying political commitment, motivating healthcare workers and health extension workers and reducing stigmatisation. These key recommendations could help in decreasing the delay in seeking health care services from appropriate institutions.

## Limitation of the study

Qualitative research is often limited by small sample sizes, making it difficult to generalize findings to a larger population. Additionally, data saturation or repeating themes may limit the novelty and originality of qualitative findings. Bias can be introduced through subjective interpretation during translation. Furthermore, because of budget constraints, this study only included health professionals working with leprosy programs, which may not fully capture the experiences of patients and community. Further larger-scale studies with larger and more diverse samples and including multiple geographical locations are warranted to validate and extend these findings.

## Conclusion

The study explored a wide range of possible challenges that affects the implementation of leprosy prevention and control strategies. Explored challenges might have an implication of delay of early detection of leprosy cases within the health facilities and the community in addition to the persistent nature of M. leprae in societies. These underscore the importance of taking action to avert those challenges by developing different strategies to improve the implementation of leprosy prevention and control program Therefore, health care providers improve the implementation of early detection, health education and community engagements by designing and developing appropriate strategies that should fit the local context..

As a result, health care providers enhance the implementation of early detection, health education, and community engagement by creating and implementing suitable strategies that align with the specific needs of the local community.

## Author Contributions

**Conceptualization:** Kebede Tefera Betru.

**Data curation:** Kebede Tefera Betru, Thuledi Makua.

**Formal analysis:** Kebede Tefera Betru, Thuledi Makua.

**Funding acquisition:** Kebede Tefera Betru.

**Investigation:** Kebede Tefera Betru, Thuledi Makua.

**Methodology:** Kebede Tefera Betru, Thuledi Makua.

**Project administration:** Kebede Tefera Betru.

**Resources:** Kebede Tefera Betru.

**Software:** Kebede Tefera Betru, Thuledi Makua.

**Supervision:** Thuledi Makua.

**Validation:** Kebede Tefera Betru, Thuledi Makua.

**Visualization:** Kebede Tefera Betru, Thuledi Makua.

**Writing – original draft:** Kebede Tefera Betru.

**Writing – review & editing:** Kebede Tefera Betru, Thuledi Makua.

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
