## [Decision Letter · Decision Letter 0]

10 Aug 2023

Dear Dr Tefera,

Thank you very much for submitting your manuscript "Challenges Experienced and Observed during the Implementation of Leprosy Strategies, Sidama Region, Southern Ethiopia: A qualitative Study" for consideration at PLOS Neglected Tropical Diseases. As with all papers reviewed by the journal, your manuscript was reviewed by members of the editorial board and by several independent reviewers. In light of the reviews (below this email), we would like to invite the resubmission of a significantly-revised version that takes into account the reviewers' comments. 

We cannot make any decision about publication until we have seen the revised manuscript and your response to the reviewers' comments. Your revised manuscript is also likely to be sent to reviewers for further evaluation.

Sincerely,

Jessica K Fairley, MD, MPH

Academic Editor

Elsio Wunder Jr

Section Editor

Reviewer's Responses to Questions

**Key Review Criteria Required for Acceptance?**

**Methods**

-Are the objectives of the study clearly articulated with a clear testable hypothesis stated?

-Is the study design appropriate to address the stated objectives?

-Is the population clearly described and appropriate for the hypothesis being tested?

-Is the sample size sufficient to ensure adequate power to address the hypothesis being tested?

-Were correct statistical analysis used to support conclusions?

-Are there concerns about ethical or regulatory requirements being met?

Reviewer #1: As an inductive, qualitative study, a hypothesis is not appropriate for the study. The study design works well for the aims of this study, which is to identify problems or barriers to prompt diagnosis and follow-up care for people with leprosy in central Ethiopia. The sample size is adequate given the method of open-ended, in-depth interviews with healthcare professionals who have expertise with leprosy.

Reviewer #2: No.

yes

no

yes

yes

no

**Results**

-Does the analysis presented match the analysis plan?

-Are the results clearly and completely presented?

-Are the figures (Tables, Images) of sufficient quality for clarity?

Reviewer #1: The analysis and presentation of results match the analysis plan. Results are well-presented, with organization of interview excerpts into themes.

Reviewer #2: yes

yes

no

**Conclusions**

-Are the conclusions supported by the data presented?

-Are the limitations of analysis clearly described?

-Do the authors discuss how these data can be helpful to advance our understanding of the topic under study?

-Is public health relevance addressed?

Reviewer #1: The conclusions are well-supported by the interview data, and the wider significance to public health is clear.

Reviewer #2: no

no

no

yes

**Editorial and Data Presentation Modifications?**

Reviewer #1: I have made some suggestions by line number here, but the author may want to have the paper reviewed by a native English speaker or work with the journal editor before publication for better clarity. There are a few minor content notes below also: 

Line 44-49: There are some issues with capitalization here and in other parts of the paper. The first letter of each of these items does not need to be capitalized. 

Line 59-60: “However the likelihood of nerve damage and 60 subsequent disability rises as the length of the diagnostic delay.” You could say, “rises with the length of the diagnostic delay” or “rises as the length of the diagnostic delay increases.” 

Line 60-61: “health professionals faces many challenges”--should be “face many challenges” 

Lines 71-73: “Armauer Hansen, a Norwegian physician who discovered the disease's bacterium in 72 the 1870s, is credited with giving leprosy, also known as Hansen's disease, its name. 73 It is a persistent infectious illness brought on by Leprae mycobacteria.”

Although it was named for him later, Hansen did not give leprosy the name Hansen’s disease but rather identified that the bacillus (Mycobacterium leprae) causes the disease. Hansen’s disease was suggested as a new name for the disease by people affected by the disease in Carville, Louisiana (United States) in the 1930s: https://www.hrsa.gov/hansens-disease/history

73-74: “It primarily spreads by nasal droplets to the mouth when there are frequent, close encounters”—should be “spreads by droplets that are emitted from the nose and mouth during breathing”

108-109: “The vast majority of them were incapable of voluntary muscle testing and feeling.”—It might be clearer to say: “The vast majority of them did not know how to conduct the diagnostic voluntary muscle tests or sensory tests for leprosy.” 

114: G2D: When you first mention this on the previous page, indicate that Grade 2 Disability is G2D with parentheses 

116-118: “The supernatural origins of traditional or religious societal beliefs about leprosy include a curse from God or an ancestor as retribution for sins, or as the result of witchcraft.” –I think you can say, “Traditional or religious beliefs” (or “Nonempirical beliefs” “about leprosy include that it is a curse from God or an ancestor, retribution from sins, or the result of witchcraft.” 

119: “ancient customs” –Since they are contemporary beliefs, I wouldn’t use the term “ancient” here, but you could give more details as well about Sidama and the influence of Christianity, Islam, etc. on beliefs about leprosy. 

169: “wide, open”—should be “open-ended” 

222: “health and safety of leprosy cases”—should be “health and safety of people affected by leprosy” 

309 and 318: “overburden works”—maybe just “an excessive work burden” 

396: “Nepal” should be “Ethiopia” 

435: “one of challenge”—should be “one of the challenges” 

480: “In this study, identified that importance”—“This study identified the importance” or “In this study, the importance of training . . . was identified.”

Reviewer #2: (No Response)

**Summary and General Comments**

Reviewer #1: This research revealed clear barriers to both early leprosy diagnosis and follow-up care for the prevention of the disabilities associated with leprosy. 

One general suggestion is to emphasize that diagnostic delay also increases the risk of leprosy reaction—one of the interviewees mentions this also. Leprosy reaction can result in people having to continue with treatment for sometimes a decade or more after MDT is complete. The author might also further emphasize how structural inequalities affect access to services in Sidama. 

A good argument for better funding and coordination of health extension worker for leprosy is that it would save money for the government over the long term. In terms of interruption in treatment, the author does a good job demonstrating, based on the interview data from health extension workers, that this is often related to factors beyond the patient’s control, such as a lack of medication or mental health or age-related disability as well as lack of resources for patient outreach. There are some excellent, concrete recommendations included in the paper as well that could be useful for national and regional health services and for international leprosy organizations that might provide funding for better support for health extension workers.

Reviewer #2: Title:

Challenges Experienced and Observed during the Implementation of Leprosy Strategies in Sidama Region, Southern Ethiopia: A Qualitative Study among Health Professionals

-Concerning the title, I suggest the authors include the approach thematic analysis and the general name for the participants as helth professionals.

Introduction:

Starting from lines 71 to 78, the authors should provide a brief background on the origin of leprosy. It is essential to mention Gerhard Armauer Hansen and his significant contribution to the discovery of the causal agent, Mycobacterium leprae.

Line 73 should be revised to replace "brought on" with "caused by" since M. leprae is the causal agent of leprosy, and this term is more commonly used.

From lines 84 to 98, the authors should update the data related to leprosy using the most recent information available, such as the Weekly Epidemiological Record (WER), 9 September 2022, Vol. 97, No 36, 2022, pp. 429–452 [EN/FR].

From lines 107 to 115, the authors should quote the total number of leprosy cases diagnosed in Ethiopia according to the latest reporting. Additionally, while mentioning the grade disability of 14%, they should clearly state that this percentage is underestimated and that expert-guided practice may reveal a higher value. They should also emphasize that MB cases represent more than 50% (1856) of new cases and are often associated with delayed diagnosis.

Methods:

A new subheading called "Type of Study and Site of Research" should be inserted. Under this heading, the authors should highlight the qualitative approach, specifically the "phenomenological approach." The period of data collection and site of research should be mentioned at the end of the sentence.

Q: Why the authors do not insert Researchers’ characteristics in this subheading highlighting experiences, qualifications, personal attributes that can improve the quality of data collection?

The authors should include information about the researchers' characteristics, their experiences, qualifications, and personal attributes that may have influenced the quality of data collection.

Study Participants and Sampling Techniques:

This section should follow a chronological order. Firstly, the authors should describe the sample size calculation used, which was based on data saturation. 

Q: Are sufficient 12 interviewers for qualitative research? Maybe this reference will help you write a paragraph pointing out the answer for this question. 

They can refer to the following source for more information: 

Hennink M, Kaiser BN. Sample sizes for saturation in qualitative research: A systematic review of empirical tests. Social science & medicine. 2022;292:114523.

Secondly, the authors should report on the purposive sampling method and the inclusion criteria for health workers at the selected and accessible health facilities who were sufficiently knowledgeable and experienced in leprosy case management and programs. They should also discuss how the participants were approached and the reasons for any refusals or dropouts.

The information regarding the site of the study (lines 165-166) should be placed before the subheading related to the site of research.

Method of Data Collection and Analysis:

Lines 168 to 183 should be rewritten to include the ethical approval by the institutional committee and the approval number. 

The authors should explain why they chose to use verbal consent instead of written informed consent, and they should provide a rationale for this choice.

The authors should also clarify who performed the interviews, why these individuals were chosen, and whether a pilot study was conducted to train the observers.

Q: Who performed the interviews? Why these persons were chosen? There was a pilot to train the observer. 

Q: Was it necessary to repeat interviews? 

Q: Were the transcripts returned to the participants to correct the mistakes? 

The instrument used for data collection (unstructured interview) should be described, along with details about the data collection site and any environmental interferences during the interviews. Information about how the data was recorded and the application used should also be included, along with the duration of the interviews and how data saturation was determined.

Regarding data analysis, line 185 should briefly explain why thematic analysis was chosen over other methods, such as grounded theory analysis. Additionally, the software used for analysis should be specified as [OpenCode 4.0 Umeå software (ICT Services and System Development and Division of Epidemiology and Global Health, Umea University, Sweden, 2015)].

Results:

The Results section should begin by introducing the main themes that emerged from data saturation.

Socio-demographic Characteristics of Study Participants:

Consider changing the title of Table 1 to include both demographic characteristics and occupational information. Add a new column to report the number of health professionals with master's degrees in various fields (nurses, physicians, psychologists, etc.). Also, include a column reporting the amount of training on leprosy.

Line 218 is incomplete and should be revised.

Challenges Experienced and Observed during the Implementation of Leprosy Strategies:

Add a section on early diagnosis challenges in leprosy, which may involve sophisticated diagnostic tests. Mention that some cases with subclinical manifestations are diagnosed by molecular and immunological tests. Discuss the importance of a multidisciplinary team for early diagnosis.

Discussion:

Line 396 should clarify why the country Nepal is mentioned in the context of Ethiopia.

After line 416, remind the readers that leprosy diagnosis is also guided by laboratory exams such as baciloscopy, serology, and molecular analysis. Positive baciloscopy can confirm MB cases.

From lines 417 to 423, provide the appropriate references. Consider discussing the differences between the ancestors of M. leprae and M. tuberculosis, their evolution, and how the diseases affect different body systems.

Conclusion:

In the conclusion section, clearly state that the challenges identified in the study can affect the implementation of leprosy prevention and control strategies. Reevaluate the statement about achieving zero autochthonous cases by 2035, considering the persistent nature of M. leprae in societies.

The limitations of this research were not stated.

The Trasnalational sense of this findings was not reported.

Overall, the revised manuscript should address these suggestions to enhance clarity, completeness, and accuracy.

PLOS authors have the option to publish the peer review history of their article (what does this mean?). If published, this will include your full peer review and any attached files.

Reviewer #1: Yes: Cassandra White

Reviewer #2: No
---

## [Decision Letter · Decision Letter 1]

2 Nov 2023

Dear Dr Tefera,

Thank you very much for submitting your manuscript "Challenges Experienced and Observed during the Implementation of Leprosy Strategies, Sidama Region, Southern Ethiopia: An Inductive Thematic Analysis of Qualitative Study among Health Professionals who Working with Leprosy Programs" for consideration at PLOS Neglected Tropical Diseases. As with all papers reviewed by the journal, your manuscript was reviewed by members of the editorial board and by several independent reviewers. The reviewers appreciated the attention to an important topic. Based on the reviews, we are likely to accept this manuscript for publication, providing that you modify the manuscript according to the review recommendations. 

The reviewer has suggested several minor details to address then we are likely to accept. Please revise according to those suggestions. 

Sincerely,

Jessica K Fairley, MD, MPH

Academic Editor

Elsio Wunder Jr

Section Editor

Reviewer's Responses to Questions

**Key Review Criteria Required for Acceptance?**

**Methods**

-Are the objectives of the study clearly articulated with a clear testable hypothesis stated?

-Is the study design appropriate to address the stated objectives?

-Is the population clearly described and appropriate for the hypothesis being tested?

-Is the sample size sufficient to ensure adequate power to address the hypothesis being tested?

-Were correct statistical analysis used to support conclusions?

-Are there concerns about ethical or regulatory requirements being met?

Reviewer #1: The authors responded to the comments in my first review. I have a few content-related comments that can be easily addressed by the authors: 

In terms of the total number of participants, was this 23 or 23 plus an additional 12 HEWs. If I’m reading this correctly, 23 potential interviewees were selected by at 12, the authors reached data saturation? 

Line 85: A more recent estimate from WHO is 3-4 million: see https://www.who.int/publications/i/item/9789290228509

Lines 604-614: This section about how morale and motivation among workers can be improved seems very generic and could instead include more specifics to the role of HEWs in Ethiopia. For example, the authors might bring back in some of the findings, such as the fact that some HEWs had other responsibilities, like tax collection, that may interfere with their work in leprosy diagnosis, or that their fears about being exposed to a person affected by leprosy may hinder their motivation.

**Results**

-Does the analysis presented match the analysis plan?

-Are the results clearly and completely presented?

-Are the figures (Tables, Images) of sufficient quality for clarity?

Reviewer #1: Yes

**Conclusions**

-Are the conclusions supported by the data presented?

-Are the limitations of analysis clearly described?

-Do the authors discuss how these data can be helpful to advance our understanding of the topic under study?

-Is public health relevance addressed?

Reviewer #1: Yes

**Editorial and Data Presentation Modifications?**

Reviewer #1: Minor editorial suggestions: 

Line 48 (in Result section of the abstract): “Importance of training related to leprosy outlined.”—I think “outlined” could be cut here and in other parts of the paper where this phrase is used. Maybe I’m misunderstanding the phrase though. Maybe the issue they are noting here is “lack of emphasis on the importance of training related to leprosy”? 

Line 50 Capitalize “strengthening” 

Line 53: Add “and” before “motivating healthcare workers” 

p. 80-81: “It is a persistent infectious illness caused by leprae mycobacteria.” You can probably cut this—already stated earlier in this paragraph

Line 150: Cut “If” here, or maybe there is a missing sentence that you can add. 

Lines 545-552: This paragraph on diagnosis is already in the Background section of the paper (lines 93-99).

**Summary and General Comments**

Reviewer #1: I think this paper contains important findings about the numerous variables that go into late diagnosis and problems with follow-up care for leprosy in a region where healthcare workers have limited opportunities for training, healthcare extension workers are pulled in different directions and may also have limited knowledge of and fears about leprosy, and where adherence to MDT is limited not only by patient behavior but by interruptions in the supply chain of medications. Issues related to the horizontalization of leprosy services, which both broadens access to MDT but involves treatment and follow-up from non-specialists, is also addressed.

PLOS authors have the option to publish the peer review history of their article (what does this mean?). If published, this will include your full peer review and any attached files.

Reviewer #1: Yes: Cassandra White

Figure Files:

Data Requirements:

Reproducibility:

References

---

## [Editor Report · Decision Letter 2]

14 Nov 2023

Dear Dr Tefera,

We are pleased to inform you that your manuscript 'Challenges Experienced and Observed during the Implementation of Leprosy Strategies, Sidama Region, Southern Ethiopia: An Inductive Thematic Analysis of Qualitative Study among Health Professionals who Working with Leprosy Programs' has been provisionally accepted for publication in PLOS Neglected Tropical Diseases.

Best regards,

Jessica K Fairley, MD, MPH

Academic Editor

Elsio Wunder Jr

Section Editor

---

## [Editor Report · Acceptance letter]

21 Nov 2023

Dear Dr Tefera,

We are delighted to inform you that your manuscript, "Challenges Experienced and Observed during the Implementation of Leprosy Strategies, Sidama Region, Southern Ethiopia: An Inductive Thematic Analysis of Qualitative Study among Health Professionals who Working with Leprosy Programs," has been formally accepted for publication in PLOS Neglected Tropical Diseases.

Best regards,

Shaden Kamhawi

co-Editor-in-Chief

Paul Brindley

co-Editor-in-Chief
